# EVOLVING COMPUTATION GRAPHS

## ABSTRACT

Graph neural networks (GNNs) have demonstrated success in modeling relational data, especially for data that exhibits homophily: when a connection between nodes tends to imply that they belong to the same class. However, while this assumption is true in many relevant situations, there are important real-world scenarios that violate this assumption, and this has spurred research into improving GNNs for these cases. In this work, we propose Evolving Computation Graphs (ECGs), a novel method for enhancing GNNs on heterophilic datasets. Our approach builds on prior theoretical insights linking node degree, high homophily, and inter vs intra-class embedding similarity by rewiring the GNNs' computation graph towards adding edges that connect nodes that are likely to be in the same class. We utilise auxiliary classifiers to identify these edges, ultimately improving GNN performance on non-homophilic data as a result. We evaluate ECGs on a diverse set of recently-proposed heterophilic datasets and demonstrate improvements over the relevant baselines. ECG presents a simple, intuitive and elegant approach for improving GNN performance on heterophilic datasets without requiring prior domain knowledge.

## 1 INTRODUCTION

Neural networks applied to graph-structured data have demonstrated success across various domains, including practical applications like drug discovery (Stokes et al., 2020; Liu et al., 2023), transportation networks (Derrow-Pinion et al., 2021), chip design (Mirhoseini et al., 2021) and theoretical advancements (Davies et al., 2021; Blundell et al., 2021). Numerous architectures fall under the category of graph neural networks (Bronstein et al., 2021), with one of the most versatile ones being Message Passing Neural Networks (Gilmer et al., 2017). The fundamental concept behind these networks is that nodes communicate with their neighbouring nodes through messages in each layer. These messages, received from neighbours, are then aggregated in a permutation-invariant manner to contribute to a new node representation.

It has been observed that the performance of graph neural networks may rely on the underlying assumption of *homophily*, which suggests that nodes are connected by edges if they are similar based on their attributes or belonging to the same class, as commonly seen in social or citation networks. However, this assumption often fails to accurately describe real-world data when the graph contains *heterophilic* edges, connecting dissimilar nodes. This observation holds particular significance since graph neural networks tend to exhibit significantly poorer performance on heterophilic graphs compared to datasets known to be homophilic. Several studies (Zhu et al., 2020b;a; Wang & Zhang, 2022; He et al., 2022) have highlighted this issue, using a mixture of strongly homophilous graphs—such as Cora, Citeseer and Pubmed (Sen et al., 2008)—as well a standard suite of six heterophilic datasets—Squirrel, Chameleon, Cornell, Texas, Wisconsin and Actor (Rozemberczki et al., 2021; Tang et al., 2009)—first curated jointly by Pei et al. (2020).

In the context of this standard suite of heterophilic graphs, it has been observed that general graph neural network architectures tend to underperform unless there is high label informativeness (Ma et al., 2021; Platonov et al., 2022). In prior work, this issue was tackled primarily by proposing modifications to the GNN architecture. These modifications include changes to the aggregation function, such as separating self- and neighbour embeddings (Zhu et al., 2020b), mixing low- and high-frequency signals (Bo et al., 2021; Luan et al., 2022), and predicting and utilising the compatibility matrix (Zhu et al., 2020c). Other approaches involve using the Jacobi basis in spectral GNNs

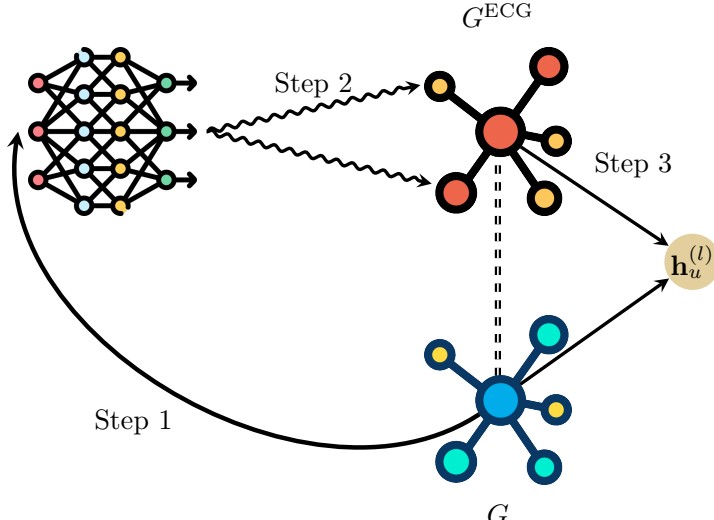

Figure 1: A simplified illustration of Evolving Computation Graphs. **Step 1:** nodes in a graph, $G$, are embedded using a pre-trained auxiliary classifier. **Step 2:** Based on these embeddings, a nearest-neighbour graph, $G^{\mathrm{EGC}}$, is generated. This graph is likely to have improved propagation and homophily properties (illustrated by similar colours between neighbouring nodes). **Step 3:** Message passing is performed, both in the original and in the ECG graph, to update node representations.

(Wang & Zhang, 2022) or learning cellular sheaves for neural sheaf diffusion (Bodnar et al., 2022) to improve performance.

However, it was recently remarked (Platonov et al., 2023) that this standard heterophilic suite has significant drawbacks, such as data originating from only three sources, two of the datasets having significant numbers of repeated nodes and improper evaluation regarding class imbalance. To address these shortcomings, a more recent benchmark suite has been introduced by Platonov et al. (2023), incorporating improvements on all of the above issues. Interestingly, once such corrections are accounted for, standard GNN models such as graph convolutional networks (Kipf & Welling, 2016, GCN), GraphSAGE (Hamilton et al., 2017, SAGE), graph attention networks (Veličković et al., 2017, GAT), and Graph Transformers (Shi et al., 2020, GT) have demonstrated superior performance compared to specialized architectures tailored specifically for heterophily—in spite of the heterophilic properties of the datasets. The notable exception is *-sep* (Zhu et al., 2020b) which has consistently improved GAT and GT by modelling self and neighbouring nodes separately.

In light of this surprising discovery, we suggest that there should be alternate routes to making the most of heterophilic datasets. Rather than attempting to modify these standard GNNs, we propose modifying their *computation graph*: effectively, enforcing messages to be sent across additional pairs of nodes. These node pairs are chosen according to a particular measure of *similarity*. If the similarity metric is favourably chosen, such a computation graph will improve the overall homophily statistics, thereby creating more favourable conditions for GNNs to perform well.

We further propose that the *modification* of the computation graph should be separate from its *utilisation*. That is, we proceed in two phases: the first phase learns the representations that allow us to construct new computation graphs, and the second phase utilises those representations to construct new computation graphs, to be utilised by a GNN in each layer. This design choice makes our method elegant, performant and easy to evaluate: the two-phase nature means we are not susceptible to bilevel optimisation (as in Franceschi et al. (2019); Kazi et al. (2022)), the graphs we use need to be precomputed exactly once rather than updated on-the-fly in every layer (as in Wang et al. (2019b)), and because the same computation graph is used across all GNN layers, we can more rigidly evaluate how useful this graph is, all other things kept equal.

Hence, the essence of our method is *Evolving Computation Graphs* (**ECG**), which uses auxiliary classifiers to generate node embeddings. These embeddings are then used to define a similarity

metric between nodes (such as cosine similarity). We then select edges in a $k$-nearest neighbour fashion: we connect each node to $k$ nodes most similar to it, according to the metric. The edges selected in this manner form a complementary graph, which we propose using in parallel with the input graph to update each node's representation. For this purpose, we use standard, off-the-shelf, GNNs. Our method is illustrated in Figure 1.

The nature of the auxiliary classifier employed in ECG is flexible and, for the purpose of this paper, we used four representative options. The first option is a point-wise MLP classifier, attempting to cluster together nodes based on the given training labels, without any graph-based biases. For the second option, we attempt the converse: utilising the given graph structure and node features, but not relying on the training labels. This is a suitable setting for a self-supervised graph representation learning method, such as BGRL (Thakoor et al., 2021), which is designed to cluster together nodes with similar local neighbourhoods—both in terms of subgraphs and features—through a bootstrapping objective (Grill et al., 2020). We may also decide to not leverage the node features in this approach, relying solely on the input graph structure to decide how to rewire it; this is the standard method utilised in prior work, and we use struc2vec (Ribeiro et al., 2017) as a representative method. Lastly, all of the above methods can be considered *weak* classifiers: they hold out either the node features, the graph structure, or the labels from the auxiliary classifier. While such decisions can make for more efficient pipelines, we also wanted to evaluate whether any empirical power is lost by doing so. Therefore, the final auxiliary classifier we attempt using for ECG is the baseline GNN itself, which doesn't hold out any information from the auxiliary classifier. In this "double-GNN" approach, two GNNs are trained: the embeddings of first one—trained over the given input graphs—are used to decide the graph structure for the second.

To evaluate the effectiveness of ECG, we conduct experiments on the benchmark suite proposed by Platonov et al. (2023). Our results demonstrate that ECG models outperform their GNN baselines in 19 out of 20 head-to-head comparisons. The most significant improvements can be noticed for GCNs—which are best suited to benefit from improved homophily—where improvements reach up to $10\%$ in absolute terms. Further, the best performing ECG models outperform a diverse set of representative heterophily-specialised GNNs, and *weak* ECG classifiers are always competitive with the full GNN classifier, allowing for a more efficient downstream pipeline.

## 2 BACKGROUND

In this section, we introduce the generic setup of learning representations on graphs, along with all of the key components that comprise our ECG method.

**Graph representation learning setup** We denote graphs by $G = (V, E)$, where $V$ is the set of nodes and $E$ is the set of edges, and we denote by $e_{uv} \in E$ the edge that connects nodes $u$ and $v$. For the datasets considered here, we can assume that the input graphs are provided to the GNNs via two inputs: the *node feature matrix*, $\mathbf{X} \in \mathbb{R}^{|V| \times k}$ (such that $\mathbf{x}_u \in \mathbb{R}^k$ are the input features of node $u \in V$), and the *adjacency matrix*, $\mathbf{A} \in \{0, 1\}^{|V| \times |V|}$, such that $a_{uv}$ indicates whether nodes $u$ and $v$ are connected by an edge. We further assume the graph is *undirected*; that is, $\mathbf{A} = \mathbf{A}^\top$. We also use $d_u = \sum_{v \in V} a_{uv} \left( = \sum_{v \in V} a_{vu} \right)$ to denote the degree of node $u$.

We focus on node classification tasks with $C$ representing the set of possible classes, where for node with input features $\mathbf{x}_u$, there is a label $y_u \in C$. Thus we aim to learn a function $f$ that minimises $\mathbb{E}[\mathcal{L}(y_u, \hat{y}_u)]$, where $\hat{y}_u$ is the prediction of $f(\mathbf{x}_u) = \hat{y}_u$, and $\mathcal{L}$ is the cross-entropy loss function.

**Graph neural networks** The one-step layer of a GNN can be summarised as follows:

$$\mathbf{h}_u^{(l)} = \phi^{(l)} \left( \mathbf{h}_u^{(l-1)}, \bigoplus_{(u,v) \in E} \psi^{(l)} \left( \mathbf{h}_u^{(l-1)}, \mathbf{h}_v^{(l-1)} \right) \right) \tag{1}$$

where, by definition, we set $\mathbf{h}_u^{(0)} = \mathbf{x}_u$. Leveraging different (potentially learnable) functions for $\phi^{(l)} : \mathbb{R}^k \times \mathbb{R}^m \to \mathbb{R}^{k'}$, $\bigoplus : \mathrm{bag}(\mathbb{R}^m) \to \mathbb{R}^m$ and $\psi^{(l)} : \mathbb{R}^k \times \mathbb{R}^k \to \mathbb{R}^m$ then recovers well-known GNN architectures. Examples include GCN (Kipf & Welling, 2016): $\psi^{(l)}(\mathbf{x}_u, \mathbf{x}_v) = \beta_{uv} \omega^{(l)}(\mathbf{x}_v)$, with $\beta_{uv} \in \mathbb{R}$ being a constant based on $\mathbf{A}$, GAT (Veličković et al., 2017): $\psi^{(l)}(\mathbf{x}_u, \mathbf{x}_v) = \alpha^{(l)}(\mathbf{x}_u, \mathbf{x}_v)\omega^{(l)}(\mathbf{x}_v)$ with $\alpha^{(l)} : \mathbb{R}^k \times \mathbb{R}^k \to \mathbb{R}$ being a (softmax-

normalised) self-attention mechanism, and *-sep* (Zhu et al., 2020b): $\phi^{(l)} = \mathbf{W}_{\text{self}}^{(l)} \phi_1^{(l)} \left( \mathbf{h}_u^{(l-1)} \right) + \mathbf{W}_{\text{agg}}^{(l)} \phi_2^{(l)} \left( \bigoplus_{(u,v) \in E} \psi^{(l)}(\mathbf{h}_u^{(l-1)}, \mathbf{h}_v^{(l-1)}) \right)$, where we explicitly decompose $\phi^{(l)}$ into two parts, with one of them ($\phi_1^{(l)}$) depending on the receiver node only.

**Homophily** has been repeatedly mentioned as an important measure of the graph, especially when it comes to GNN performance. Intuitively, it corresponds to an assumption that neighbouring nodes tend to share labels: $a_{uv} = 1 \implies y_u = y_v$, which is often the case for many industrially-relevant real world graphs (such as social networks). Intuitively, a graph with high homophily will make it easier to exploit neighbourhood structure to derive more accurate node labels.

However, in spite of the importance of quantifying homophily in a graph, there is no universally-agreed-upon metric for this. One very popular metric, used by several studies, is *edge homophily* Abu-El-Haija et al. (2019), which measures the proportion of homophilic edges:

$$\texttt{h-edge} = \frac{|(u,v) \in E : y_u = y_v|}{|E|} \tag{2}$$

while Platonov et al. (2022) also introduces *adjusted homophily* to account for number of classes and their distributions:

$$\texttt{h-adj} = \frac{\texttt{h-edge} - \sum_{k=1}^{C} D_k^2 / (2|E|)^2}{1 - \sum_{k=1}^{C} D_k^2 / (2|E|)^2} \tag{3}$$

where $D_k = \sum_{u:y_u=k} d_u$, the sum of degrees for the nodes belonging to class $k$.

Additionally, the *label informativeness* (LI) measure proposed in Platonov et al. (2022) measures how much information about a node's label is gained by observing its neighbour's label, on average. It is defined as

$$\texttt{LI} = I(y_\xi, y_\eta) / H(y_\xi) \tag{4}$$

where $(\xi, \eta) \in E$ is a uniformly-sampled edge, $H$ is the Shannon entropy and $I$ is mutual information.

**Auxiliary classifier** In order to derive novel computation graphs which are likely to result in higher test performance, we likely require "novel" homophilic connections to emerge. To do this, we leverage several auxiliary classifier options.

**MLPs** Arguably the simplest way to make a weak-but-efficient classifier, as above, is to withhold access to the graph structure ($\mathbf{A}$), and force the model to classify the nodes in pure isolation from one another. This is effectively a standard multi-layer perceptron (MLP) applied pointwise. Another way of understanding this model is setting $\mathbf{A} = \mathbf{I}_{|V|}$, or equivalently, $E = \{(u,u) \mid u \in V\}$, in Equation 1, which is sometimes referred to as the Deep Sets model (Zaheer et al., 2017). We train this model by using cross-entropy against the training nodes' labels ($\mathbf{y}_{\text{tr}}$) and, once trained, use the final layer activations, $\mathbf{h}_u^{(L)}$—for a model with $L$ layers—as our MLP embeddings.

**BGRL** While using the embeddings from an MLP can offer a solid way to improve homophily metrics, their confidence will degrade for nodes where the model is less accurate outside of the training set—arguably, the nodes we would like to improve predictions on the most. Accordingly, as a converse approach, we may withhold access to the training labels ($\mathbf{y}_{\text{tr}}$). Now the model is forced to arrange the node representations in a way that will be mindful of the input features and graph structure, but without knowing the task specifics upfront, and hence not vulnerable to overfitting on the training nodes. Such a classifier naturally lends itself to self-supervised learning on graphs.

Bootstrapped graph latents (Thakoor et al., 2021, BGRL) is a state-of-the-art self-supervised graph representation learning method based on BYOL (Grill et al., 2020). BGRL learns two GNN encoders with identical architecture; an *online* encoder, $\mathcal{E}_\theta$, and a *target* encoder, $\mathcal{E}_\phi$. BGRL also contains a *predictor* network $p_\theta$. We offer a "bird's eye" view of how BGRL is trained, and defer to Thakoor et al. (2021) for implementation details.

At each step of training, BGRL proceeds as follows. First, two data augmentations (e.g. random node and edge dropout) are applied to the input graph, obtaining augmented graphs $(\mathbf{X}_1, \mathbf{A}_1)$ and $(\mathbf{X}_2, \mathbf{A}_2)$. Then, the two encoders are applied to these augmentations, recovering a pair of latent node embeddings: $\mathbf{H}_1 = \mathcal{E}_\theta(\mathbf{X}_1, \mathbf{A}_1)$, $\mathbf{H}_2 = \mathcal{E}_\phi(\mathbf{X}_2, \mathbf{A}_2)$. The first embedding is additionally

passed through the predictor network: $\mathbf{Z}_1 = p_\theta(\mathbf{H}_1)$. At this point, BGRL attempts to preserve the cosine similarity between all the corresponding nodes in $\mathbf{Z}_1$ and $\mathbf{H}_2$, via the following loss function:

$$\mathcal{L}_{\mathrm{BGRL}} = -\frac{\mathbf{Z}_1 \mathbf{H}_2^\top}{\|\mathbf{Z}_1\| \|\mathbf{H}_2\|} \tag{5}$$

The parameters of the online encoder $\mathcal{E}_\theta$ and predictor $p_\theta$ are updated via SGD on $\mathcal{L}_{\mathrm{BGRL}}$, and the parameters of the target encoder $\mathcal{E}_\phi$ are updated as the EMA of the online encoder's parameters. Once the training procedure concludes, typically only the online network $\mathcal{E}_\theta$ is retained, and hence the embeddings $\mathbf{H} = \mathcal{E}_\theta(\mathbf{X}, \mathbf{A})$ are the BGRL embeddings of the input graph given by node features $\mathbf{X}$ and adjacency matrix $\mathbf{A}$.

Owing to its bootstrapped objective, BGRL does not require the generation of negative samples, and is hence computationally efficient compared to contrastive learning approaches. Further, it is very successful at large scales; it was shown by Addanki et al. (2021) that BGRL's benefits persist on industrially relevant graphs of hundreds of millions of nodes, leading to one of the top-3 winning entries at the OGB-LSC competition (Hu et al., 2021). This is why we employ it as a representative self-supervised embedding method for our ECG framework.

**struc2vec**  We may decide to weaken the BGRL approach one step further by withholding access to the node features ($\mathbf{X}$). This is now a pure graph rewiring method, and we choose to use struc2vec (Ribeiro et al., 2017) for this purpose, as it is the preferred method of choice in various prior works.

**(Double) GNN**  All previous methods are weak classifiers—the model does not have access to inputs ($\mathbf{X}$), graph structure ($\mathbf{A}$) and training labels ($\mathbf{y}_{\mathrm{tr}}$) simultaneously. It is our opinion that such methods should be competitive, as they would discover novel kinds of edges to draw under more challenging training settings, rather than amplifying the homophily already present in $\mathbf{A}$. To validate this, we also compare against a variant of ECG that has access to all of the above, i.e., an ECG where a GNN is used as an auxiliary classifier. We make the choice of GNN match the downstream GNN model which will use the auxiliary edges. This auxiliary classifier can also be seen as a variant of the MLP where we do not set $\mathbf{A} = \mathbf{I}_{|V|}$, but retain the original adjacency of the input graph.

## 3  EVOLVING COMPUTATION GRAPHS

Armed with the concepts above, we are now ready to describe the steps of the ECG methodology. Please refer to Algorithm 1 for a pseudocode summary.

**Step 1: Embedding extraction**  Firstly, we assume that an appropriate auxiliary classifier has already been trained (as discussed in previous sections), and is capable of producing node embeddings. We start by invoking this classifier to obtain ECG embeddings $\mathbf{H}_{\mathrm{ECG}} = \gamma(\mathbf{X}, \mathbf{A})$. We study four simple but potent variants of $\gamma$, as per the previous section:

**MLP:** In this case, we utilise a simple deep MLP[1]; that is, $\gamma(\mathbf{X}, \mathbf{A}) = \sigma\left(\sigma\left(\mathbf{X}\mathbf{W}_1\right)\mathbf{W}_2\right)$, where $\mathbf{W}.$ are the MLP's weights, and $\sigma$ is the GELU activation (Hendrycks & Gimpel, 2016).

**BGRL:** In this case, we set $\gamma = \mathcal{E}_\theta$, the online encoder of BGRL. For our experiments, we utilise a publicly available off-the-shelf implementation of BGRL provided by DGL[2] (Wang et al., 2019a), which uses a two-layer GCN (Kipf & Welling, 2016) as the base encoder.

**struc2vec:** In this case, we make $\gamma$ a look-up table, directly retrieving for each node its embeddings as precomputed from struc2vec (Ribeiro et al., 2017).

**GNN:** In this case, we make $\gamma$ have the same form as the GNN which will utilise the ECG edges; that is, $\gamma$ has the form in Equation 8, but with separate parameters.

The parameters of $\gamma$ are kept frozen throughout what follows, and are not to be further trained on.

---

[1]Note that this MLP, as well as the GNN described later, only compute high-dimensional embeddings of each node; while training $\gamma$, an additional logistic regression layer is attached to their architecture.

[2]https://github.com/dmlc/dgl/tree/master/examples/pytorch/bgrl

**Step 2: Graph construction**   Having obtained $\mathbf{H}_{\text{ECG}}$, we can now use it to compute a similarity metric between the nodes, such as cosine similarity, as follows:

$$\mathbf{S} = \mathbf{H}_{\text{ECG}}\mathbf{H}_{\text{ECG}}^{\top} \qquad \hat{s}_{uv} = \frac{s_{uv}}{\|\mathbf{h}_{\text{ECG}_u}\|\|\mathbf{h}_{\text{ECG}_v}\|} \tag{6}$$

Based on this similarity metric, for each node $u \in V$ we select its neighbourhood $\mathcal{N}_u^{\text{ECG}}$ to be its $k$ nearest neighbours in $\mathbf{S}$ (where $k$ is a tunable hyperparameter):

$$\mathcal{N}_u^{\text{ECG}} = \text{top-}k_{v \in V}\,\hat{s}_{uv} \tag{7}$$

Equivalently, we construct a new computation graph, $G^{\text{ECG}} = (V, E^{\text{ECG}})$, with edges $E^{\text{ECG}} = \{(u,v) \mid u \in V \wedge v \in \mathcal{N}_u\}$. These edges are effectively determined by the auxiliary classifier.

**Step 3: Parallel message passing**   Finally, once the ECG graph, $G^{\text{ECG}}$, is available, we can run our GNN of choice over it. To retain the topological benefits contained in the input graph structure, we opt to run two GNN layers in parallel—one over the input graph (as in Equation 1), and one over the ECG graph, as follows:

$$\mathbf{h}_{\text{INP}_u}^{(l)} = \phi_{\text{INP}}^{(l)}\left(\mathbf{h}_u^{(l-1)}, \bigoplus_{(u,v)\in E} \psi_{\text{INP}}^{(l)}\left(\mathbf{h}_u^{(l-1)}, \mathbf{h}_v^{(l-1)}\right)\right) \tag{8}$$

$$\mathbf{h}_{\text{ECG}_u}^{(l)} = \phi_{\text{ECG}}^{(l)}\left(\mathbf{h}_u^{(l-1)}, \bigoplus_{(u,v)\in E^{\text{ECG}}} \psi_{\text{ECG}}^{(l)}\left(\mathbf{h}_u^{(l-1)}, \mathbf{h}_v^{(l-1)}\right)\right) \tag{9}$$

Then the representation after $l$ layers is obtained by jointly transforming these two representations:

$$\mathbf{h}_u^{(l)} = \mathbf{W}^{(l)}\mathbf{h}_{\text{INP}_u}^{(l)} + \mathbf{U}^{(l)}\mathbf{h}_{\text{ECG}_u}^{(l)} \tag{10}$$

where $\mathbf{W}^{(l)}$ and $\mathbf{U}^{(l)}$ are learnable parameters.

Equations 8–10 can then be repeatedly iterated, much like is the case for any standard GNN layer. As $G^{\text{ECG}}$ may contain noisy edges which do not contribute to useful propagation of messages, we apply DropEdge (Rong et al., 2020) when propagating over the ECG graph, with probability $p_{de} = 0.5$.

## 4   EXPERIMENTS

We evaluate the performance of ECG on five heterophilic datasets, recently-proposed by Platonov et al. (2023): `roman-empire`, `amazon-ratings`, `minesweeper`, `tolokers` and `questions`. All five datasets are node classification tasks, testing for varying levels of homophily in the input (`roman-empire` has the highest label informativeness), different connectivity profiles (`tolokers` is the most dense, `questions` has the lowest values of clustering coefficients) and providing both real-world (`amazon-ratings`), and synthetic (`minesweeper`) datasets.

We ran ECG as an extension on standard GNN models, choosing the "-sep" variant (Zhu et al., 2020b) for GAT and GT as it was noted to improve their performance consistently on these tasks (Platonov et al., 2023). Thus, our baselines are GCN, GraphSAGE, GAT-sep and GT-sep, which we extend by modifying their computation graph as presented in Section 3. For each ECG model, we ran several variants, depending on which auxiliary classifier was used to select $E^{\text{ECG}}$.

For each of these architectures, the hyper-parameters to sweep are the number of neighbours sampled in the ECG graph, $k$ (selected from $\{3, 10, 20\}$), the edge dropout rate used on it (selected from $\{0., 0.5\}$, the hidden dimension of the graph neural networks, where the one ran on the original graph $G$ always matches the one ran on $G^{\text{ECG}}$ (selected from $\{256, 512\}$), as well as the standard choice of number of layers (selected from $\{2, 3, 4, 5\}$). In Appendix A, we present additional information on the experiments, together with the hyper-parameters corresponding to the best validation results.

In Table 1, we show the test performance corresponding to the highest validation score among all embedding possibilites for each of the five datasets and for each of the four baselines. Altogether,

---

**Algorithm 1:** Evolving Computation Graph for Graph Neural Networks: ECG-GNN

---

**Input:** Graph $G = (V, E)$; Node Feature Matrix $\mathbf{X}$; Adjacency Matrix $\mathbf{A}$.
**Hyper-parameters:** Value of $k$; Drop edge probability $p_{de}$; Number of layers $L$;
**Output:** Predicted labels $\hat{\mathbf{y}}$
**begin**

```
/* Step 1:  Extract embeddings */
```
$\mathbf{H}_{\mathrm{ECG}} \leftarrow \gamma(\mathbf{X}, \mathbf{A})$                               `/* Embeddings stored in matrix */`
```
/* Step 2:  Construct ECG graph */
```
$\mathbf{S} \leftarrow \mathbf{H}_{\mathrm{ECG}}\mathbf{H}_{\mathrm{ECG}}^{\top}$
**for** $u \in V$ **do**
    **for** $v \in V$ **do**
        $\hat{s}_{uv} \leftarrow s_{uv}/(\|\mathbf{h}_{\mathrm{ECG}_u}\|\|\mathbf{h}_{\mathrm{ECG}_v}\|)$          `/* Compute cosine similarities */`
    $\mathcal{N}_u^{\mathrm{ECG}} \leftarrow \text{top-}k_{v \in V}\hat{s}_{uv}$            `/* Compute k nearest neighbours of u */`
$E^{\mathrm{ECG}} \leftarrow \{(u,v) \,|\, u \in V \wedge v \in \mathcal{N}_u\}$            `/* Construct the ECG edges */`
```
/* Step 3:  Parallel message passing on G and G^ECG */
```
**for** $u \in V$ **do**
    $\mathbf{h}_u^0 \leftarrow \mathbf{x}_u$                           `/* Setting initial node features */`
**for** $l \leftarrow 1$ **to** $L$ **do**
    `/* Message passing propagation on G and G^ECG respectively */`
    $E_{(l)}^{\mathrm{ECG}} \leftarrow \mathrm{DropEdge}(E^{\mathrm{ECG}}, p_{de})$          `/* Randomly drop edges in G^ECG */`
    **for** $u \in V$ **do**
        $\mathbf{h}_{\mathrm{INP}_u}^{(l)} \leftarrow \phi_{\mathrm{INP}}^{(l)}\left(\mathbf{h}_u^{(l-1)}, \bigoplus_{(u,v) \in E} \psi_{\mathrm{INP}}^{(l)}\left(\mathbf{h}_u^{(l-1)}, \mathbf{h}_v^{(l-1)}\right)\right)$        `/* on G */`
        $\mathbf{h}_{\mathrm{ECG}_u}^{(l)} \leftarrow \phi_{\mathrm{ECG}}^{(l)}\left(\mathbf{h}_u^{(l-1)}, \bigoplus_{(u,v) \in E_{(l)}^{\mathrm{ECG}}} \psi_{\mathrm{ECG}}^{(l)}\left(\mathbf{h}_u^{(l-1)}, \mathbf{h}_v^{(l-1)}\right)\right)$     `/* on G^ECG */`
        $\mathbf{h}_u^{(l)} \leftarrow \mathbf{W}^{(l)}\mathbf{h}_{\mathrm{INP}_u}^{(l)} + \mathbf{U}^{(l)}\mathbf{h}_{\mathrm{ECG}_u}^{(l)}$ `/* Updating the node representation */`
```
/* Predict node labels */
```
**for** $u \in V$ **do**
    $p_u \leftarrow \mathrm{softmax}(\mathbf{W}^{(c)}\mathbf{h}_u^{(L)}) \ \hat{y}_u \leftarrow \arg\max_{c \in C} p_c$     `/* Predicted class label */`
**return** $\hat{\mathbf{y}}$

---

there are 20 dataset-model combinations that ECG is tested on. We find that on 19 out of these 20 combinations (marked with arrow up in the table), using ECG improves the performance of the corresponding GNN architecture, the only exception being GraphSAGE on `amazon-ratings`.

Moreover, we observe the highest gains in performance are achieved by ECG-GCN, being as high as $10.84\%$ (absolute values), in a manner that is correlated with the homophily of the dataset. This confirms the hypothesis that, due to the aggregation function it employs, GCN is also the architecture most prone to performance changes based on the homophily of the input graph.

Through the detailed ablations on ECG types in Appendix A, we note that the "double GNN" approach never significantly outperforms weak auxiliary classifiers. This not only highlights that weaker classifiers are already potent at capturing relevant information for drawing auxiliary edges, but also that it is possible to deploy ECG using a highly efficient pre-training pipeline.

### 4.1 QUALITATIVE STUDIES

In Table 2, we analyse the properties of the complementary graphs $G^{\mathrm{ECG}}$ with $k = 3$ nearest neighbours. We note that this represents the graph used by ECG-GCN, which preferred lower values of $k$, while the optimal values of $k$ for SAGE, GAT-sep and GT-sep were on the higher end, varying between 3, 10 and 20 depending on the dataset.

We observe that MLP-ECG confirms our hypothesis: taking the edges corresponding to the pairs of nodes marked as most similar by the MLP results in a graph $G^{\mathrm{ECG}}$ with high homophily, especially compared to the original input graph. It is important to note that all of our MLP-ECG graphs were obtained with a relatively shallow MLP, which, as it can be seen in Table 1 lacks in performance compared to the graph-based methods. However, our method's success in conjunction with

Table 1: Benchmarking ECG on Platonov et al. (2023). We report accuracy for `roman-empire` and `amazon-ratings` and ROC AUC for `minesweeper`, `tolokers`, and `questions`.

| Model | roman-empire | amazon-ratings | minesweeper | tolokers | questions |
|---|---|---|---|---|---|
| MLP | $65.88_{\pm0.38}$ | $45.90_{\pm0.52}$ | $50.89_{\pm1.39}$ | $72.95_{\pm1.06}$ | $70.34_{\pm0.76}$ |
| GCN | $73.69_{\pm0.74}$ | $49.13_{\pm0.37}$ | $89.75_{\pm0.52}$ | $83.64_{\pm0.67}$ | $77.13_{\pm0.31}$ |
| ECG-GCN | $84.53_{\pm0.26}$ ($\uparrow$) | $51.12_{\pm0.38}$ ($\uparrow$) | $92.89_{\pm0.10}$ ($\uparrow$) | $\mathbf{84.91}_{\pm0.14}$ ($\uparrow$) | $77.50_{\pm0.35}$ ($\uparrow$) |
| SAGE | $85.74_{\pm0.67}$ | $\mathbf{54.41}_{\pm0.29}$ | $93.66_{\pm0.13}$ | $82.56_{\pm0.10}$ | $76.44_{\pm0.62}$ |
| ECG-SAGE | $87.88_{\pm0.25}$ ($\uparrow$) | $53.45_{\pm0.27}$ ($\downarrow$) | $94.11_{\pm0.07}$ ($\uparrow$) | $83.81_{\pm0.12}$ ($\uparrow$) | $77.43_{\pm0.25}$ ($\uparrow$) |
| GAT-sep | $88.75_{\pm0.41}$ | $53.44_{\pm0.34}$ | $93.91_{\pm0.35}$ | $83.78_{\pm0.43}$ | $76.79_{\pm0.71}$ |
| ECG-GAT-sep | $\mathbf{89.62}_{\pm0.18}$ ($\uparrow$) | $53.65_{\pm0.39}$ ($\uparrow$) | $\mathbf{94.52}_{\pm0.20}$ ($\uparrow$) | $84.92_{\pm0.16}$ ($\uparrow$) | $77.67_{\pm0.18}$ ($\uparrow$) |
| GT-sep | $87.32_{\pm0.39}$ | $52.18_{\pm0.80}$ | $92.29_{\pm0.47}$ | $82.52_{\pm0.92}$ | $78.05_{\pm0.93}$ |
| ECG-GT-sep | $89.56_{\pm0.16}$ ($\uparrow$) | $53.25_{\pm0.39}$ ($\uparrow$) | $93.69_{\pm0.34}$ ($\uparrow$) | $84.28_{\pm0.38}$ ($\uparrow$) | $78.23_{\pm0.26}$ ($\uparrow$) |
| $H_2$GCN | $60.11_{\pm0.52}$ | $36.47_{\pm0.23}$ | $89.71_{\pm0.31}$ | $73.35_{\pm1.01}$ | $63.59_{\pm1.46}$ |
| CPGNN | $63.96_{\pm0.62}$ | $39.79_{\pm0.77}$ | $52.03_{\pm5.46}$ | $73.36_{\pm1.01}$ | $65.96_{\pm1.95}$ |
| GPR-GNN | $64.85_{\pm0.27}$ | $44.88_{\pm0.34}$ | $86.24_{\pm0.61}$ | $72.94_{\pm0.97}$ | $55.48_{\pm0.91}$ |
| FSGNN | $79.92_{\pm0.56}$ | $52.74_{\pm0.83}$ | $90.08_{\pm0.70}$ | $82.76_{\pm0.61}$ | $\mathbf{78.86}_{\pm0.92}$ |
| GloGNN | $59.63_{\pm0.69}$ | $36.89_{\pm0.14}$ | $51.08_{\pm1.23}$ | $73.39_{\pm1.17}$ | $65.74_{\pm1.19}$ |
| FAGCN | $65.22_{\pm0.56}$ | $44.12_{\pm0.30}$ | $88.17_{\pm0.73}$ | $77.75_{\pm1.05}$ | $77.24_{\pm1.26}$ |
| GBK-GNN | $74.57_{\pm0.47}$ | $45.98_{\pm0.71}$ | $90.85_{\pm0.58}$ | $81.01_{\pm0.67}$ | $74.47_{\pm0.86}$ |
| JacobiConv | $71.14_{\pm0.42}$ | $43.55_{\pm0.48}$ | $89.66_{\pm0.40}$ | $68.66_{\pm0.65}$ | $73.88_{\pm1.16}$ |

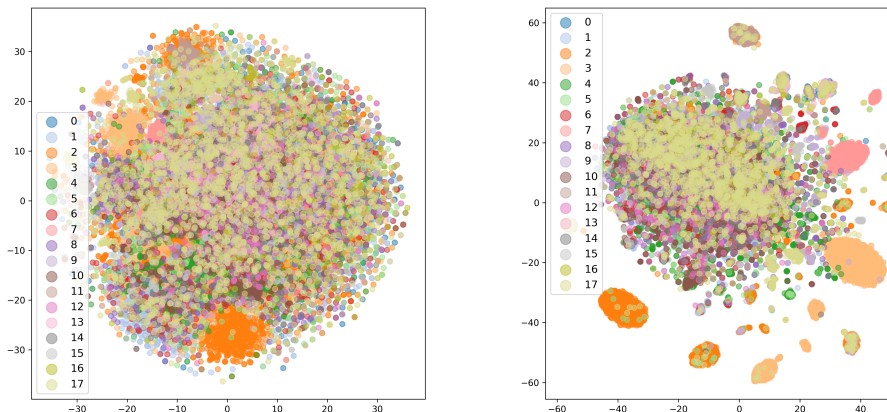

Figure 2: For `roman-empire`, we use a random GCN layer to obtain node embeddings based on the original graph $G$ (left) or the ECG graph $G^{\mathrm{ECG}}$ (right). The colours correspond to the ground-truth labels of the nodes. See Appendix B for the version of this figure with a trained (ECG-)GCN.

GNNs shows that even a weak classifier can be used to generate homophilic graphs that improve performance when complementing information provided by the input data. Table 5 further includes statistics of the graphs obtained from BGRL embeddings and MLP and BGRL together.

In Figure 2, we also verify how predictive of the node classes the graph topology is when obtained from the original data compared to when we build a complementary graph $G^{\mathrm{ECG}}$. More precisely, we first build the graph $G^{\mathrm{ECG}}$ as presented in Step 1 of Algorithm 1, using pre-trained MLP embeddings. Then we use a randomly initialised GCN to compute node embeddings on the input graph $G$, as well as on $G^{\mathrm{ECG}}$. We visualise these two sets of node embeddings using t-SNE (Van der Maaten & Hinton, 2008) by projecting to a 2D space, attributing the colour of each point based

Table 2: Statistics of the original heterophilic graphs, and of the ECG obtained from MLP.

|  | roman-empire | amazon-ratings | minesweeper | tolokers | questions |
|---|---|---|---|---|---|
| edges | $32,927$ | $93,050$ | $39,402$ | $519,000$ | $153,540$ |
| edge homophily | $0.05$ | $0.38$ | $0.68$ | $0.59$ | $0.84$ |
| adjusted homophily | $-0.05$ | $0.14$ | $0.01$ | $0.09$ | $0.02$ |
| LI | $0.11$ | $0.04$ | $0.00$ | $0.01$ | $0.00$ |
| ECG ($k = 3$) edges | $67,986$ | $73,476$ | $30,000$ | $35,274$ | $146,763$ |
| MLP-ECG edge homophily | $0.73$ | $0.66$ | $0.79$ | $0.79$ | $0.97$ |
| MLP-ECG adjusted homophily | $0.7$ | $0.53$ | $0.33$ | $0.4$ | $0.41$ |
| MLP-ECG LI | $0.65$ | $0.33$ | $0.16$ | $0.19$ | $0.28$ |

on the node's ground truth label. We can observe that using the $G^{\mathrm{ECG}}$ topology leads to more distinguishable clusters corresponding to the classes even without any training, thus supporting the enhancements in performance when building ECG-GNN.

## 5 RELATED WORK

Many specialised architectures have been proposed to tackle GNNs limitations in modeling heterophilic data. H$_2$GCN (Zhu et al., 2020b) separates ego and neighbour embeddings, uses higher-order neighbourhoods and combines representations from intermediate layers. CPGNN (Zhu et al., 2020a) learns a compatibility matrix to explicitly integrate information about label correlations and uses it to modulate messages. Similarly, GGCN (Yan et al., 2021) modifies GCN through degree corrections and signed messages, based on an insight linking heterophily and oversmoothing. FAGCN (Bo et al., 2021) uses a self-gating mechanism to adaptively integrate low-frequency signals, high-frequency signals and raw features and ACM-GNN (Luan et al., 2022) extends it to enable adaptive channel mixing node-wise. GPRGNN (Chien et al., 2021) learns Generalized PageRank weights that adjust to node label patterns. FSGNN (Maurya et al., 2022) proposes Feature Selection GNN which separates node feature aggregation from the depth of the GNN through multiplication the node features with different powers and transformations of the adjacency matrix and uses a softmax to select the relevant features. GloGNN (Li et al., 2022) leverages global nodes to aggreagte information, while GBK-GNN (Du et al., 2022) uses bi-kernel feature transformation.

Most relevant to ECG-GNN could be GeomGCN (Pei et al., 2020) and the work of Suresh et al. (2021). The former uses network embedding methods to find neighbouring nodes in the corresponding latent space, to be then aggregated with the neighbours in the original graph, over which a two-level aggregation is then performed. Similarly, (Suresh et al., 2021) modifies the computation graph by computing similarity of degree sequences for different numbers of hops. However, in both cases, the input node features and labels, which could provide useful signal, are not used.

However, Platonov et al. (2023) points that the standard datasets on which these models were tested, such as Squirrel, Chameleon, Cornell, Texas and Wisconsin, had considerable drawbacks: high number of duplicated nodes, highly imbalanced classes and lack of diversity in setups considered. When evaluated on the newly proposed heterophilic suite, most specialised architectures are outperformed by their standard counter-parts such as GCN, SAGE, GAT and GT, with only the separation of ego and neighbour embeddings from Zhu et al. (2020b) maintaining an advantage.

## 6 CONCLUSIONS

We present Evolving Computation Graphs for graph neural networks, ECG-GNN. This is a two-phase method focused on improving the performance of standard GNNs on heterophilic data. Firstly, it builds an evolved computation graph, formed from the original input and a complementary set of edges determined by an auxiliary classifier. Then, the two components of this computation graph are modelled in parallel by two GNNs processors, and projected to the same embeddings space after each propagation layer. This simple and elegant extension of existing graph neural networks proves to be very effective – for four models considered on five diverse heterophilic datasets, the ECG-GNN enhances the performance in $95\%$ of the cases.

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

## A  HYPERPARAMETER SETTINGS AND DETAILED ABLATION RESULTS

We used the same experimental setup as presented in Platonov et al. (2023). Results are aggregated over ten random splits of the data, with each run taking $50\%$ of the nodes for training, $25\%$ for validation, and $25\%$ for testing. The following hyperparameters are tuned for all models and baselines, using the average validation performance across the splits:

- Number of layers, $L \in \{1, 2, 3, 4, 5\}$.
- The dimensionality for the baseline latent embeddings were $d \in \{512, 1024\}$ and for the ECG variants $d \in \{256, 512\}$, to account for the additional parameters incurred by ECG's two parallel processing streams.

Additionally, for the ECG models only, the following hyperparameters were swept:

- Embeddings used by ECG, $\tau \in \{\text{MLP}, \text{BGRL}, \text{struc2vec}, \text{GNN}, \text{MLPBGRL}, \text{MLP} \to \text{GNN}\}$, referring to:

  *MLP:*  Using the embeddings from a pre-trained MLP;

  *BGRL:*  Using the embeddings from a pre-trained BGRL model;

  *struc2vec:*  Using the embeddings from a struc2vec model;

  *GNN:*  Using the embeddings from a pre-trained GNN;

  *MLPBGRL:*  Using the normalised concatenation of the MLP and BGRL embeddings;

  *MLP→GNN:*  Using the embeddings from a pre-trained MLP-ECG model of the same type.

  > Note that, for methods requiring access to labels (such as MLP or GNN), a separate set of embeddings is computed for every dataset split (to avoid test data contamination). For self-supervised methods like BGRL or struc2vec, no labels are used, and hence a single set of embeddings is produced for all experiments.

- The number of neighbours sampled per node, $k \in \{3, 10, 20\}$.
- The DropEdge rate, $p_{de} \in \{0.0, 0.5\}$.

The model configuration with the best-performing average validation performance is then evaluated on the corresponding test splits, producing the aggregated performances reported in Table 1.

The best-performing hyperparameters for each model type on each dataset are given in Table 3. Each individual experiment has been executed on a single NVIDIA Tesla P100 GPU, and the longest training time allocated to an individual experiment has been six hours (on the `questions` dataset).

For convenience, and to assess the relative benefits of various ECG embedding sources, we provide in Table 4 an expanded version of 1, showing the test performance obtained by the tuned version of each ECG variant, for every embedding type.

For additional information, the anonymised code can be found at `https://anonymous.4open.science/r/evolving_computation_graphs-97B7/`.

## B  ADDITIONAL QUALITATIVE RESULTS OF ECG

In Figure 3 we provide visualisations of the node embeddings on `roman-empire`, comparing those obtained by a trained GCN to those obtained from ECG-GCN. Similarly to the figure in the main text, we observe a more structured latent space in the latter, aligned with better empirical performance when classifying.

In Table 5, we provide the homophily analysis of the original graphs, as well as the graphs obtained using the embeddings of a pre-trained MLP, a pre-trained BGRL and from leveraging the two models together.

Table 3: The best-performing hyperparameters for each GNN propagation rule in our experiments. The only experiment where the baseline model outperforms ECG is the SAGE propagation layer on `amazon-ratings`; hence, the hyperparameters $k$ and $p_{de}$ are irrelevant.

|  | roman-empire | amazon-ratings | minesweeper | tolokers | questions |
|---|---|---|---|---|---|
| **MLP** | | | | | |
| $L$ | 2 | 1 | 5 | 5 | 1 |
| $d$ | 512 | 512 | 512 | 512 | 512 |
| **GCN** | | | | | |
| $L$ | 5 | 2 | 4 | 5 | 3 |
| $d$ | 512 | 512 | 256 | 512 | 256 |
| $\tau$ | MLPBGRL | MLP→GNN | GNN | GNN | BGRL |
| $k$ | 3 | 3 | 3 | 3 | 3 |
| $p_{de}$ | 0.5 | 0.5 | 0.5 | 0.0 | 0.0 |
| **SAGE** | | | | | |
| $L$ | 5 | 2 | 5 | 4 | 5 |
| $d$ | 512 | 1024 | 256 | 256 | 512 |
| $\tau$ | BGRL | Baseline | BGRL | struc2vec | struc2vec |
| $k$ | 10 | — | 20 | 10 | 10 |
| $p_{de}$ | 0.5 | — | 0.5 | 0.5 | 0.0 |
| **GAT-sep** | | | | | |
| $L$ | 5 | 2 | 5 | 4 | 4 |
| $d$ | 512 | 512 | 256 | 256 | 512 |
| $\tau$ | BGRL | MLP→GNN | MLP | struc2vec | struc2vec |
| $k$ | 10 | 3 | 20 | 10 | 10 |
| $p_{de}$ | 0.5 | 0.5 | 0.5 | 0.5 | 0.0 |
| **GT-sep** | | | | | |
| $L$ | 5 | 2 | 5 | 5 | 5 |
| $d$ | 512 | 512 | 256 | 256 | 512 |
| $\tau$ | BGRL | MLP→GNN | GNN | struc2vec | struc2vec |
| $k$ | 20 | 3 | 3 | 20 | 20 |
| $p_{de}$ | 0.5 | 0.5 | 0.5 | 0.5 | 0.5 |

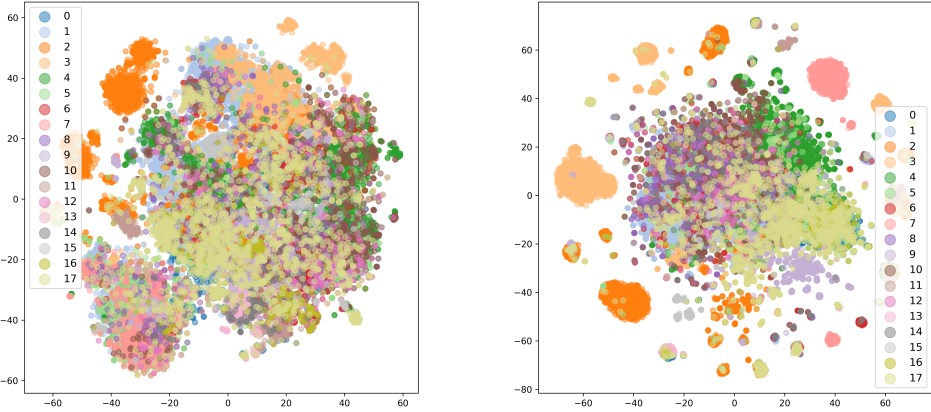

Figure 3: For `roman-empire`, we use a pre-trained GCN (left) and ECG-GCN (right) layer to obtain node embeddings. The colours correspond to the ground-truth labels of the nodes.

Table 4: Detailed breakdown of model performance on the datasets proposed by Platonov et al. (2023). MLP, GCN, SAGE, GAT-sep and GT-sep are the baselines, while all the other models are variants of ECG. Red marks the best performance on each dataset for each of the considered GNN architectures and the corresponding ECGs. Accuracy is reported for `roman-empire` and `amazon-ratings`, and ROC AUC is reported for `minesweeper`, `tolokers`, and `questions`.

| | roman-empire | amazon-ratings | minesweeper | tolokers | questions |
|---|---|---|---|---|---|
| MLP | $65.88 \pm 0.38$ | $45.90 \pm 0.52$ | $50.89 \pm 1.39$ | $72.95 \pm 1.06$ | $70.34 \pm 0.76$ |
| GCN | $73.69 \pm 0.74$ | $49.13 \pm 0.37$ | $89.75 \pm 0.52$ | $83.64 \pm 0.67$ | $77.13 \pm 0.31$ |
| MLP-ECG-GCN | $83.55 \pm 0.39$ | $50.99 \pm 0.64$ | $92.63 \pm 0.10$ | $84.81 \pm 0.25$ | $76.25 \pm 0.59$ |
| BGRL-ECG-GCN | $80.59 \pm 0.48$ | $48.99 \pm 0.28$ | $92.35 \pm 0.10$ | $84.25 \pm 0.22$ | $77.50 \pm 0.35$ |
| Struc2vec-ECG-GCN | $81.62 \pm 0.31$ | $49.18 \pm 0.27$ | $92.44 \pm 0.09$ | $84.22 \pm 0.13$ | $76.50 \pm 0.30$ |
| GNN-ECG-GCN | $84.52 \pm 0.41$ | $48.95 \pm 0.30$ | $92.89 \pm 0.10$ | $84.91 \pm 0.14$ | $77.30 \pm 0.28$ |
| MLPBGRL-ECG-GCN | $84.53 \pm 0.26$ | $50.11 \pm 0.60$ | $92.47 \pm 0.50$ | $84.73 \pm 0.23$ | $77.32 \pm 0.31$ |
| MLP→GNN-ECG-GCN | $84.39 \pm 0.22$ | $51.12 \pm 0.38$ | $92.56 \pm 0.23$ | $84.35 \pm 0.31$ | $75.16 \pm 0.87$ |
| SAGE | $85.74 \pm 0.67$ | $54.41 \pm 0.29$ | $93.66 \pm 0.13$ | $82.56 \pm 0.10$ | $76.44 \pm 0.62$ |
| MLP-ECG-SAGE | $85.82 \pm 0.62$ | $53.32 \pm 0.39$ | $94.10 \pm 0.08$ | $82.60 \pm 0.23$ | $76.13 \pm 0.41$ |
| BGRL-ECG-SAGE | $87.88 \pm 0.25$ | $53.12 \pm 0.32$ | $94.11 \pm 0.07$ | $82.61 \pm 0.29$ | $77.23 \pm 0.36$ |
| Struc2vec-ECG-SAGE | $87.67 \pm 0.28$ | $53.40 \pm 0.45$ | $94.10 \pm 0.09$ | $83.81 \pm 0.12$ | $77.43 \pm 0.25$ |
| GNN-ECG-SAGE | $87.05 \pm 0.24$ | $53.36 \pm 0.60$ | $94.09 \pm 0.07$ | $82.38 \pm 0.27$ | $75.83 \pm 1.00$ |
| MLPBGRL-ECG-SAGE | $86.50 \pm 0.34$ | $52.34 \pm 0.92$ | $94.01 \pm 0.07$ | $82.55 \pm 0.18$ | $76.55 \pm 0.33$ |
| MLP→GNN-ECG-SAGE | $85.94 \pm 0.57$ | $53.45 \pm 0.27$ | $93.77 \pm 0.12$ | $82.52 \pm 0.22$ | $75.53 \pm 0.64$ |
| GAT-sep | $88.75 \pm 0.41$ | $53.44 \pm 0.34$ | $93.91 \pm 0.35$ | $83.78 \pm 0.43$ | $76.79 \pm 0.71$ |
| MLP-ECG-GAT-sep | $88.22 \pm 0.36$ | $52.98 \pm 0.30$ | $94.52 \pm 0.20$ | $83.91 \pm 0.32$ | $77.30 \pm 0.47$ |
| BGRL-ECG-GAT-sep | $89.62 \pm 0.18$ | $52.20 \pm 0.57$ | $94.24 \pm 0.15$ | $84.23 \pm 0.25$ | $77.38 \pm 0.18$ |
| Struc2vec-ECG-GAT-sep | $89.48 \pm 0.19$ | $53.28 \pm 0.45$ | $94.37 \pm 0.14$ | $84.92 \pm 0.16$ | $77.67 \pm 0.18$ |
| GNN-ECG-GAT-sep | $88.64 \pm 0.21$ | $53.08 \pm 0.48$ | $94.46 \pm 0.12$ | $83.90 \pm 0.19$ | $76.35 \pm 0.60$ |
| MLPBGRL-GAT-sep | $88.73 \pm 0.37$ | $51.06 \pm 0.73$ | $94.39 \pm 0.20$ | $84.11 \pm 0.23$ | $76.97 \pm 0.45$ |
| MLP→GNN-ECG-GAT-sep | $88.04 \pm 0.32$ | $53.65 \pm 0.39$ | $93.97 \pm 0.19$ | $83.75 \pm 0.30$ | $75.61 \pm 0.74$ |
| GT-sep | $87.32 \pm 0.39$ | $52.18 \pm 0.80$ | $92.29 \pm 0.47$ | $82.52 \pm 0.92$ | $78.05 \pm 0.93$ |
| MLP-ECG-GT-sep | $88.56 \pm 0.35$ | $52.68 \pm 0.65$ | $93.62 \pm 0.27$ | $83.65 \pm 0.29$ | $77.82 \pm 0.43$ |
| BGRL-ECG-GT-sep | $89.56 \pm 0.16$ | $52.37 \pm 0.30$ | $93.55 \pm 0.18$ | $82.97 \pm 0.26$ | $78.12 \pm 0.32$ |
| Struc2vec-ECG-GT-sep | $89.07 \pm 0.23$ | $52.43 \pm 0.49$ | $93.66 \pm 0.23$ | $84.28 \pm 0.38$ | $78.23 \pm 0.26$ |
| GNN-ECG-GT-sep | $89.16 \pm 0.23$ | $52.71 \pm 0.40$ | $93.69 \pm 0.34$ | $83.43 \pm 0.28$ | $77.71 \pm 0.43$ |
| MLPBGRL-GT-sep | $88.70 \pm 0.30$ | $52.29 \pm 0.60$ | $93.52 \pm 0.25$ | $84.00 \pm 0.24$ | $77.85 \pm 0.45$ |
| MLP→GNN-ECG-GT-sep | $88.62 \pm 0.46$ | $53.25 \pm 0.39$ | $92.69 \pm 0.34$ | $83.41 \pm 0.44$ | $75.50 \pm 1.13$ |

Table 5: Statistics of the original heterophilous graphs and of the evolutionary computation graph obtained from MLP, BGRL and MLPBGRL.

| | roman-empire | amazon-ratings | minesweeper | tolokers | questions |
|---|---|---|---|---|---|
| edges | $32,927$ | $93,050$ | $39,402$ | $519,000$ | $153,540$ |
| edge homophily | 0.05 | 0.38 | 0.68 | 0.59 | 0.84 |
| adjusted homophily | $-0.05$ | 0.14 | 0.01 | 0.09 | 0.02 |
| LI | 0.11 | 0.04 | 0.00 | 0.01 | 0.00 |
| ECG($k=3$) edges | $67,986$ | $73,476$ | $30,000$ | $35,274$ | $146,763$ |
| MLP-ECG edge homophily | 0.73 | 0.66 | 0.79 | 0.79 | 0.97 |
| MLP-ECG adjusted homophily | 0.7 | 0.53 | 0.33 | 0.4 | 0.41 |
| MLP-ECG LI | 0.65 | 0.33 | 0.16 | 0.19 | 0.28 |
| BGRL-ECG edge homophily | 0.16 | 0.3 | 0.68 | 0.6 | 0.93 |
| BGRL-ECG adjusted homophily | 0.06 | 0.02 | 0.12 | 0.08 | 0.01 |
| BGRL-ECG LI | 0.1 | 0.03 | 0.05 | 0.03 | 0.03 |
| MLPBGRL-ECG edge homophily | 0.72 | 0.56 | 0.63 | 0.79 | 0.98 |
| MLPBGRL-ECG adjusted homophily | 0.69 | 0.38 | 0.1 | 0.03 | 0.5 |
| MLPBGRL-ECG LI | 0.62 | 0.23 | 0.03 | 0.06 | 0.4 |

