# OpenReview forum: "Evolving Computation Graphs"
_ICLR.cc/2024/Conference — ICLR 2024 Conference Withdrawn Submission_

### Official Review · Reviewer_gyLE · 2023-10-31

**Soundness:** 3 good
**Presentation:** 3 good
**Contribution:** 3 good
**Rating:** 6
**Confidence:** 4

**Summary:**

Important poin: To first clear the context, the paper uses a new benchmark dataset evaluation for heterophily proposed by Platonov et al 2023[A].

Review--
Unlike existing works, the paper does not  attempt to modify  standard GNNs to tackle heterophily. They  focus on  modifying the computation graph itself. The idea is to have two phases. First phase, to learn a model(could be MLP, GNN etc.) that  learns the representations of nodes which allows  constructing new computation graphs. Phase 2 ) Utilizing  representations oh phase 1 to consstruct new computation graphs for the GNN. Their proposal of modifying the computation graph enforces messages to be sent across additional pairs of nodes based upon similarity of nodes( from phase 1).

The paper is easy to follow. Pseudocode etc. is provided. Empirical performance is mostly superior.  The authors also provide visualizations to enhance better understanding of their approach..


[A]Oleg Platonov, Denis Kuznedelev, Michael Diskin, Artem Babenko, and Liudmila Prokhorenkova.
A critical look at the evaluation of gnns under heterophily: are we really making progress? arXiv
preprint arXiv:2302.11640, 2023

**Strengths:**

1. Presentation is good. The paper is easy to follow.

2. The approach seems to be novel to the best of my knowledge.( Atleast for the problem in hand)

3.  The authors additionally visualize the obtained node embeddings based upon original graph and new graphs. This enhances clarity of work.

4. The experiment evaluation is good.

5. The authors evaluate on datasets with varying degree of homophily. The performance of the proposed model is pretty good on varying degree of homophily. Further, the authors use a more recent( and relevant) benchmark dataset.

6. The authors also release their codebase.

Significance:
The paper is simple yet effective. The ideas proposed in the paper could further accelerate research in this direction.

**Weaknesses:**

1. In page 2, when the authors say "propose modifying their computation graph".  I believe the context w.r.t the heterophilic graphs is somewhat missing in this. Could the authors motivate this  better w.r.t the heterophily graphs?

2. For MLP in page4, it is not clear whether node feature X is used or not.

**Questions:**

1. What does it mean by a randomly initialized GCN in Page 8 last para  which is relevant for Fig.2? Request the authors to motivate this choice in the revision.  Could the authors cite any similar work showing embedding of random initialized GCN?

2. It might be a good idea to move Table 2 before Table 1 to maintain continuity.

Request to answer the points in weakness section.


I believe answering above questions could improve the manuscript.

---

### Official Review · Reviewer_3tzc · 2023-10-31

**Soundness:** 3 good
**Presentation:** 2 fair
**Contribution:** 1 poor
**Rating:** 3
**Confidence:** 4

**Summary:**

The paper is proposing an approach to tackle heterophily problem in GNNs. The basic idea can be outlined as follows:
1] Train an embedding model using either MLP, GNN, BGRL or some other weak model which does not have access to at least one of graph, features or labels
2] Use these embeddings to reconstruct the graph using kNN
3] Learn the final model using the constructed graph and any GNN model

**Strengths:**

1. A very simple approach.
1. The results look very promising.

**Weaknesses:**

1. **Incremental novelty:** Similar ideas have been tried for example in [A], although the paper does not explicitly call out heterophily, but they working in the setting of noisy graphs and heterophily can be considered a noisy graphs where all the edges connecting nodes of different classes as noise.
1. **Missing Baselines:** The first point brings us to the next, [A] proposes a more enhanced version which learns the graph using bi-level optimization and such methods should have been considered as potential baselines for this proposed method. In fact, there are several follow ups to this work [B, C, D]. Strangely, [A] is part of the citation, but there is no discussion related to [A] anywhere in the write up.
1. **Baseline Hyperparameter Tuning:** The baseline numbers are not looking very reliable and there is question as to whether these baselines have been tuned properly. Not tuning baselines correctly has been found to be an issue in the past [E, F]. In fact, similar comments have been raised on the paper where the datasets considered in this paper were proposed [G]. Just as an example, with some amount of tuning, we were able to get 71.97 for GPRGNN model on roman-empire, which is higher than the reported 64.85.

* [A] Learning Discrete Structures for Graph Neural Networks, ICML 2019.
* [B] Iterative Deep Graph Learning for Graph Neural Networks: Better and Robust Node Embeddings, NeurIPS 2020.
* [C] GLAM: Graph Learning by Modeling Affinity to Labeled Nodes for Graph Neural Networks, ICLR 2021.
* [D] A Survey on Graph Structure Learning: Progress and Opportunities, arXiv 2022.
* [E] Pitfalls of Graph Neural Network Evaluation, NeurIPS 2018.
* [F] Pitfalls in Evaluating GNNs under Label Poisoning Attacks, ICLR 2023.
* [G] https://openreview.net/forum?id=tJbbQfw-5wv

**Questions:**

1. How were the initial embeddings models trained, particularly the ones that used training labels? Did they use validation labels for hyper-parameter tuning? Unfortunately, the shared code loads these embeddings directly from data directly, so it is not clear where these initial embeddings generation was done and what setting is used.

---

### Official Review · Reviewer_7f7X · 2023-11-01

**Soundness:** 2 fair
**Presentation:** 3 good
**Contribution:** 2 fair
**Rating:** 5
**Confidence:** 4

**Summary:**

The paper addresses the problem of learning on heterophilous graphs. The method is the following: first, train an auxiliary model that produces node embeddings. Then, construct a k-NN graph on these embeddings. Finally, train a GNN model on both the original and new graphs. As an auxiliary model, six variants are considered: MLP which does not use the graph structure; BGRL which does not use the node labels; struc2vec which uses only the graph structure; GNN which uses all the available information, MLPBGRL that concatenates MLP and BGRL embeddings; and MLP→GNN that is the proposed MLP-ECG used as an auxiliary model.

**Strengths:**

1. The proposed method is simple and easy to implement.
2. The paper is clearly written and easy to follow.
3. The improved performance is achieved on the recently proposed heterophilous benchmark.

**Weaknesses:**

1. **Comparison with existing methods.** There are many rewiring methods in the literature, but there is no comparison with such approaches. In particular, some methods like Suresh et al. (2021) consider the similarity based on the graph structure. Thus, they are similar to the struc2vec variant of EGC. According to Tables 3 and 4, struc2vec is competitive and often outperforms other approaches.

2. **Motivation.** The main motivation for graph rewiring is improved homophily For instance, it is written in the introduction "If the similarity metric is favourably chosen, such a computation graph will improve the overall homophily statistics, thereby creating more favourable conditions for GNNs to perform well." However, among the considered auxiliary models, only MLP clearly creates homophilous graphs. This is supported by Table 5, where BGRL-ECG has low homophily. Thus, some considered auxiliary models do not improve homophily but still work well, while MLP is rarely the best auxiliary model.

3. **Complexity overhead.** The proposed approach is significantly more complex since it requires additional models to be trained and more budget for parameter tuning. In Table 1, the chosen auxiliary model is a hyperparameter, so all of them need to be implemented and compared. Moreover, while not mentioned in the main text, some complex auxiliary models are used: MLPBGRL and MLP→GNN.

**Questions:**

Q1. How does the proposed method relate to Suresh et al. (2021) and other graph rewiring methods (in terms of the general approach)?

Q2. Figure 2 (ECG) is shown only for the MLP auxiliary model which by definition aims to cluster nodes with the same label together. How does this figure look for other auxiliary models?

Q3. Table 4 shows that the best auxiliary model can be different. A discussion about which model is better and whether it is sufficient to use one of them would be helpful.

Q4. Table 5: what are the values for other considered auxiliary models? Do some of them lead to high homophily?

---

### Official Review · Reviewer_gESK · 2023-11-03

**Soundness:** 2 fair
**Presentation:** 2 fair
**Contribution:** 2 fair
**Rating:** 3
**Confidence:** 5

**Summary:**

This paper introduces a method, ECG, for reconstructing structures to create more homophilic graphs. Four embedding methods, each with an auxiliary classifier, are proposed for learning node representation. Graph structures are then reconstructed based on the top-k feature similarities. Parallel aggregations on both the original and new graphs are designed to update node features.

**Strengths:**

1. The auxiliary classifier is interesting and useful as evaluated in the experiments.

2. This paper is easy to follow and understand.

**Weaknesses:**

1. The main contribution is unclear.

It appears that the main contribution is the framework design as shown in Fig.1. However, existing methods such as WRGAT [1] also use a similar pipeline where graph construction and learning are separated into two stages. The four embedding extraction methods used in step 1 are intriguing, but the paper only introduces them without justifying their advantages and disadvantages compared to each other and methods in the literature.


2. The evaluations of the proposed method are insufficient.

	- Hyper-parameters: There are many settings that could affect performance, such as embedding methods and neighbor number k, which are treated as hyper-parameters. Particularly for the embedding methods, it seems that the performance reported in Table 1 is selected based on test performance. Table 4 reports the test performance of ECGs with different embedding methods, and the highest ones are selected and reported in Table 1. If the authors treat the embedding methods as hyper-parameters, they should select the appropriate choice with the **validation performance**. If they are not hyper-parameters, the authors should justify why these choices are selected.


	- Performance analysis: The performance analysis is conducted based on Table 1. However, based on Table 4, lower performance compared to vanilla GNN methods can be observed, indicating the insufficient expressiveness of the designed method.

	- Baseline selection: This paper focuses on reconstructing graph structures, and related methods such as Geom-GCN and WAGAT[1] should be considered as baselines, similar to the usage of H2GCN. The statement "However, in both cases, the input node features and labels, which could provide useful signal, are not used." needs clarification.

	- Limited datasets: Five datasets are insufficient to evaluate the effectiveness of ECGs. The large-scale datasets mentioned in [2] should be considered.






[1] Breaking the Limit of Graph Neural Networks by Improving the Assortativity of Graphs with Local Mixing Patterns. KDD 21

[2] Large Scale Learning on Non-Homophilous Graphs: New Benchmarks and Strong Simple Methods. NeurIPS 2021

**Questions:**

Please check the weaknesses.